# North Sea region energy system towards 2050: integrated offshore grid and sector coupling drive offshore wind installations

Matti Koivisto[1], Juan Gea-Bermúdez[1,2], Polyneikis Kanellas[1], Kaushik Das[1], Poul Sørensen[1]

[1]Department of Wind Energy, Technical University of Denmark, Roskilde, Denmark
[2]Department of Management, Technical University of Denmark, Lyngby, Denmark

*Correspondence to*: Matti Koivisto (mkoi@dtu.dk)

**Abstract.** This paper analyses several energy system scenarios towards 2050 for the North Sea region. With focus on offshore wind power, the impacts of meshed offshore grid and sector coupling are studied. First, a project-based scenario, where each offshore wind power plant is connected individually to onshore, is compared to a meshed grid scenario. Both the amount of offshore wind installed, and the level of curtailment are assessed. Then, these results are compared to a scenario with sector coupling included. The results show that while the introduction of a meshed grid can increase the amount of offshore wind installed towards 2050, sector coupling is expected to be a more important driver for increasing offshore wind installations. In addition, sector coupling can significantly decrease the level of offshore wind curtailment.

## 1 Introduction

The North Sea offers high offshore wind power potential. In addition, several existing and planned transmission lines are located in the region. Consequently, a meshed offshore grid in the North Sea has been proposed as an option for connecting transmission and offshore wind generation investments in the region (Konstantelos et al., 2017), (Gorenstein Dedecca et al., 2017), (de Decker et al., 2011). The meshed offshore grid can be optimised assuming a fixed onshore generation development (Gorenstein Dedecca et al., 2017), (Gorenstein Dedecca et al., 2018); however, joint optimisation of the onshore and offshore parts has been suggested to find the overall optimal system (Gea-Bermudez et al., 2020), (Gorenstein Dedecca & Hakvoort, 2016). This paper presents results from comparing an integrated approach, where meshed grid and onshore transmission and generation investments are optimised jointly, to a project-based scenario, where each offshore wind power plant (OWPP) is connected individually to onshore.

Another development that can have significant impact on variable renewable energy (VRE) generation is sector coupling (Brown et al., 2018), (Münster et al., 2020), (WindEurope, 2018). With expected increase in electricity consumption, there is more load that can be met by VRE generation. In addition, sector coupling can provide additional flexibility to the power system, e.g., via electrification of heating demand in both individual heating (Brown et al., 2018) and industry (Gea-Bermudez, Koivisto, et al., 2019). The value of cross-border and cross-sector coupling were compared in (Thellufsen & Lund, 2017), with

cross-sector coupling showing stronger benefits. However, the complementarity of the two was not analysed. In addition, the geographical resolution was low. In (Brown et al., 2018), a combined optimisation of sector coupling and transmission reinforcement was carried out, considering individual-user heating sector, road transport and power-to-gas. The results show that sector coupling and transmission expansion reduce costs, with a combination of the two found optimal. Energy supply for all of Europe for one year was optimised on hourly resolution, with one node per country and without including industry. The

benefits of coupling the electricity and transport sectors were shown in (Helgeson & Peter, 2020). (Hedegaard et al., 2012) showed that electric vehicles (EVs) can help in integrating more wind energy to the power system. Different types of storages were compared in (Victoria et al., 2019), with the result that large-scale thermal storage can help in balancing the system at a seasonal level, while EVs contribute to balancing in the short-term.

In this paper, the North Sea region energy system is analysed on regional level for the Nordic countries (matching the Nord Pool market bidding zones). Regional modelling is applied also for Germany to model important intra-country grid congestions. Transmission expansion is modelled jointly with sector coupling to study their combined impact. When considering sector coupling, the electricity and heating sectors are optimized jointly towards 2050, with electrification of industry and district heating expansion also modelled. In addition, increasing EV penetration is considered. Electrification of

industry heat demand is modelled considering three temperature levels. Electrification increases electricity consumption; however, sector coupling has also potential to provide flexibility to the system, which is modelled. Following the modelling in (Gea-Bermudez et al., 2020), intertemporal rather than myopic optimisation is carried out to capture the long-term perspective in investment decisions.

With focus on the effects on offshore wind power, this paper analyses and compares the impacts of a meshed North Sea offshore grid and sector coupling. Both the expected installation of offshore wind towards 2050 and the level of curtailment are analysed and compared. The analyses are carried out using a combination of CorRES (Correlations in Renewable Energy Sources) and Balmorel tools. CorRES provides the wind and solar generation time series used in analysing the impacts of VRE generation on the energy system. The expected technology development of VRE generation towards 2050 is modelled and

offshore wind installations are analysed considering nearshore and far offshore AC and DC investments, as shown in (Koivisto, Gea-Bermudez, et al., 2019). Balmorel takes the CorRES simulations as an input and analyses the expected evolution of the North Sea region energy system towards 2050. With simulated operation of the energy system, considering both the electricity and heating sectors, Balmorel is used to model the behaviour of the system on hourly level.

The project-based and the meshed offshore grid scenarios have been published before (Koivisto, Gea-Bermudez, et al., 2019). However, they are supplemented with recent results on the level of VRE curtailment (Gea-Bermudez, Das, et al., 2019). The presented scenario with sector coupling is new work. In addition to presenting the scenario with sector coupling modelled, this

paper contributes by comparing the expected impacts of introducing a meshed offshore grid in the North Sea to the impacts of sector coupling both on the amount of offshore wind power installed and the level of curtailment expected.


The paper is structured as follows. Section 2 describes the methodology used in analysing the North Sea region energy system development towards 2050. Section 3 presents the results for the studied scenarios and compares them. Section 4 provides discussion on the results and assumptions. Section 5 provides a conclusion of the presented results.

## 2 Methodology

All scenarios are analysed using a combination of CorRES and Balmorel, following the approach shown in (Gea-Bermudez et al., 2020), with CorRES providing the VRE time series and Balmorel carrying out the energy system modelling, as shown in Figure 1. The following subsections present both tools. Figure 2 shows the North Sea region countries in focus in this paper.

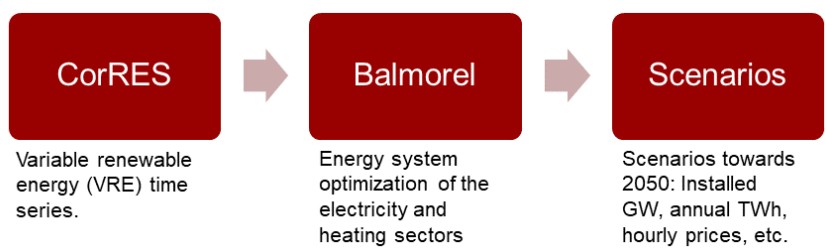

**Figure 1. The scenario modelling flow chart.**

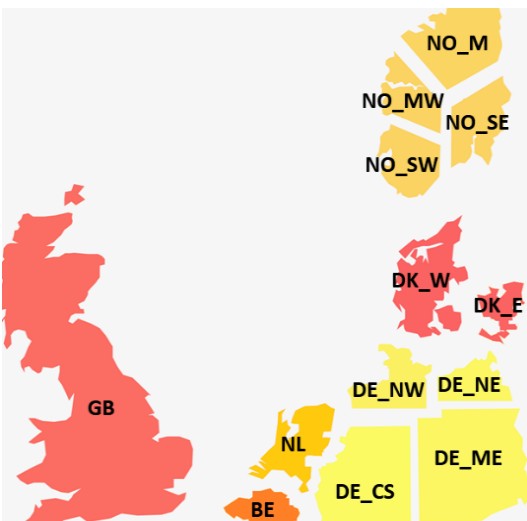

**Figure 2. The North Sea region countries in focus (the regional split refers to the project-based and meshed grid scenarios); figure is taken from (Koivisto, Gea-Bermudez, et al., 2019). The north region of NO (NO_N) is not shown in the graph but included in the aggregated results.**

### 2.1 CorRES

CorRES (Koivisto, Das, et al., 2019) is used for simulating the VRE generation time series used in Balmorel. CorRES allows modelling of pan-European scale wind and solar PV generation time series (Nuño et al., 2018), with both the spatial (between the modelled countries and regions) and temporal dependencies in VRE generation modelled. In addition to analysing current VRE installation, CorRES can be used in analysing the expected impacts of technology development on both the capacity factors (CFs) and the spatiotemporal dependencies in the VRE time series (Koivisto, Maule, et al., 2019). For the analysed scenarios, wind power is expected to experience both increased hub heights and lower specific power towards 2050. The expected technology developments are linked to the costs of VRE installations, as shown in (Gea-Bermudez et al., 2020), to model the combination of both costs decreasing and technology advancing. For offshore wind, the distance from shore impacts CF and the cost of grid connection, with both nearshore and far offshore AC and DC installations modelled, as presented in (Koivisto, Gea-Bermudez, et al., 2019). For onshore wind and solar PV, each region is split in at least two resource grade areas with different potentials and CFs (Gea-Bermudez, Koivisto, et al., 2019). This is done to model that VRE resource quality is not constant within a region.

### 2.2 Balmorel

For energy system optimisation, the Balmorel model ([www.balmorel.com](www.balmorel.com)) (Wiese et al., 2018) is used. Balmorel is open source ([github.com/balmorelcommunity/Balmorel](github.com/balmorelcommunity/Balmorel)), deterministic and takes a bottom-up approach. The objective function in Balmorel is to minimize total system costs (Wiese et al., 2018). Balmorel has been traditionally used to perform joint optimisation of the electricity and district heating sectors, although it is being constantly developed to include additional sectors, e.g., industry (included in this paper), individual heating (not included) and transport (partially included with EV scenarios based on (Gea-Bermudez, Koivisto, et al., 2019)). Joint modelling of the electricity and heating sectors allows assessment of benefits from integrating the markets of the different sectors. The setup of the model is similar to (Gea-Bermudez, Koivisto, et al., 2019), although with some important differences. The main difference is that the modelling of industry in Balmorel is based in this paper on three temperature levels (low (<100°C), medium, and high (>500°C)) to reflect that not all technologies can satisfy all types of heat demand. Heat pumps are assumed to be capable of satisfying low temperature demand, CHP low and medium temperature demand, and boilers and electrification low, medium, and high temperature demand.

Compared to (Gea-Bermudez, Koivisto, et al., 2019), the only tax and tariff used in this paper is the $CO_2$ tax, which pushes VRE penetration on the expense of fossil generation. Assumed $CO_2$ tax values are: 30, 90 and 120 €$_{2015}$/ton in 2025, 2035, and 2045, respectively; they are based on (NORDEN and IEA, 2016). Since the costs of biofuels are very sensitive to their demand, biofuels are modelled with stepwise price functions as in (Gea-Bermudez, Koivisto, et al., 2019). The aggregated biofuel

potentials and corresponding prices are shown in Table 1 for the countries in focus. More details about the assumptions and technologies included in the model can be found in (Gea-Bermudez, Koivisto, et al., 2019).


In this paper, Balmorel is used to perform for the sector coupling scenario: 1) a capacity development optimization; and 2) day-ahead market simulations. In (Gea-Bermudez, Koivisto, et al., 2019) only the first optimization was performed. Investments in generation, storage, power transmission and district heating expansion, as well as decommissioning of generation capacity, are allowed. Due to computational complexity, 8 spread-over-the-year weeks with 1-every-3 hours are
used as representative time steps in the optimization. VRE time series are scaled using the approach described in (Gea-Bermudez et al., 2020), so the statistical representation of the full year is kept. Unit commitment integer variables are relaxed in this optimization. EV charging is assumed to be non-flexible. The capacity development is then used as input in the day-ahead market simulations.

The day-ahead market simulation for the sector coupling scenario has two steps: 1) full year simulations to obtain storage levels at the beginning of each day, planned maintenance and daily resource allocation; and 2) day-by-day market simulation. Resource allocation is relevant for limited fuels, such as municipal solid waste or biomass. In the full year simulations, all days and 1 every 3 hours are used, EV charging is assumed to be non-flexible, and the relaxation of unit commitment integer variables is applied due to computational limitations. The method is based, and further explained in (Gea-Bermudez, Das, et
al., 2019). In the day-by-day simulation, EV smart charging is allowed. The hourly dispatch values from the day-by-day market simulations are used to compute annual generation, demand, emissions, prices, intercountry transmission flows and wind power curtailment, among others.

**Table 1. Aggregated biofuel potentials and corresponding prices for the countries in focus (the UK, NL, DK, BE, DE and NO); each fuel is modelled with a stepwise price function with three levels. Data gathered as shown in (Gea-Bermudez, Koivisto, et al., 2019).**

| Fuel | Biogas | | | Straw | | | Wood chips | | | Wood pellets | | |
|---|---|---|---|---|---|---|---|---|---|---|---|---|
| Price ($€_{2012}$/GJ) | 9.5 | 16.5 | 142.3 | 3.5 | 6.0 | 51.9 | 4.1 | 7.1 | 61.0 | 5.8 | 10.1 | 87.1 |
| Potential (PJ) | 7 | 202 | 530 | 39 | 1086 | 2857 | 253 | 6965 | 18313 | 39 | 1077 | 2831 |

## 3 Results

This section presents and compares the resulting scenarios. The first subsection compares the meshed offshore grid scenario to the project-based one. First, the renewable energy shares and offshore wind installations are compared. Then, the expected levels of VRE curtailment are assessed. In subsection 3.2, the scenario with sector coupling is presented, considering the
renewable energy share, annual energy generation mix, amount of offshore wind installations and the expected level of curtailment. The sector coupling scenario does not include a meshed offshore grid.

The project-based and offshore grid scenarios were performed optimizing investments in GB, DK, NO, DE, BE, and NL (Figure 2), whereas the capacity development for surrounding countries was exogenously given (Koivisto, Gea-Bermudez, et al., 2019). In the sector coupling scenario, the capacity development was optimized in all included countries. Additionally, compared to the project-based and meshed offshore grid set up, in the sector coupling scenario UK was analysed instead of GB, a different regional set up for DE was defined to better capture transmission congestion, and Estonia, Lithuania and Latvia were excluded from the runs to reduce computational complexity. However, the scenarios can still be compared on aggregate level, as is done in the following subsections. The results are shown for the countries in focus (Figure 2), so the results between all analysed scenarios can be compared.

## 3.1 Impacts of a meshed offshore grid

This section compares the project-based and the meshed offshore grid scenario. The scenarios have been presented before (Koivisto, Gea-Bermudez, et al., 2019); however, the second subsection adds additional information regarding VRE curtailment. The main difference between the scenarios can be seen in Figure 3: in the project-based scenario, only country-to-country transmission lines are allowed (OWPPs are connected to shore project-by-project); in the meshed offshore grid scenario, meshed connections in the North Sea are allowed in Balmorel investment optimisation (in addition, OWPPs can be connected to the hubs that are part of the meshed offshore grid infrastructure). More information on how the meshed offshore gird is modelled in the Balmorel investment optimisation can be found in (Gea-Bermudez et al., 2020).

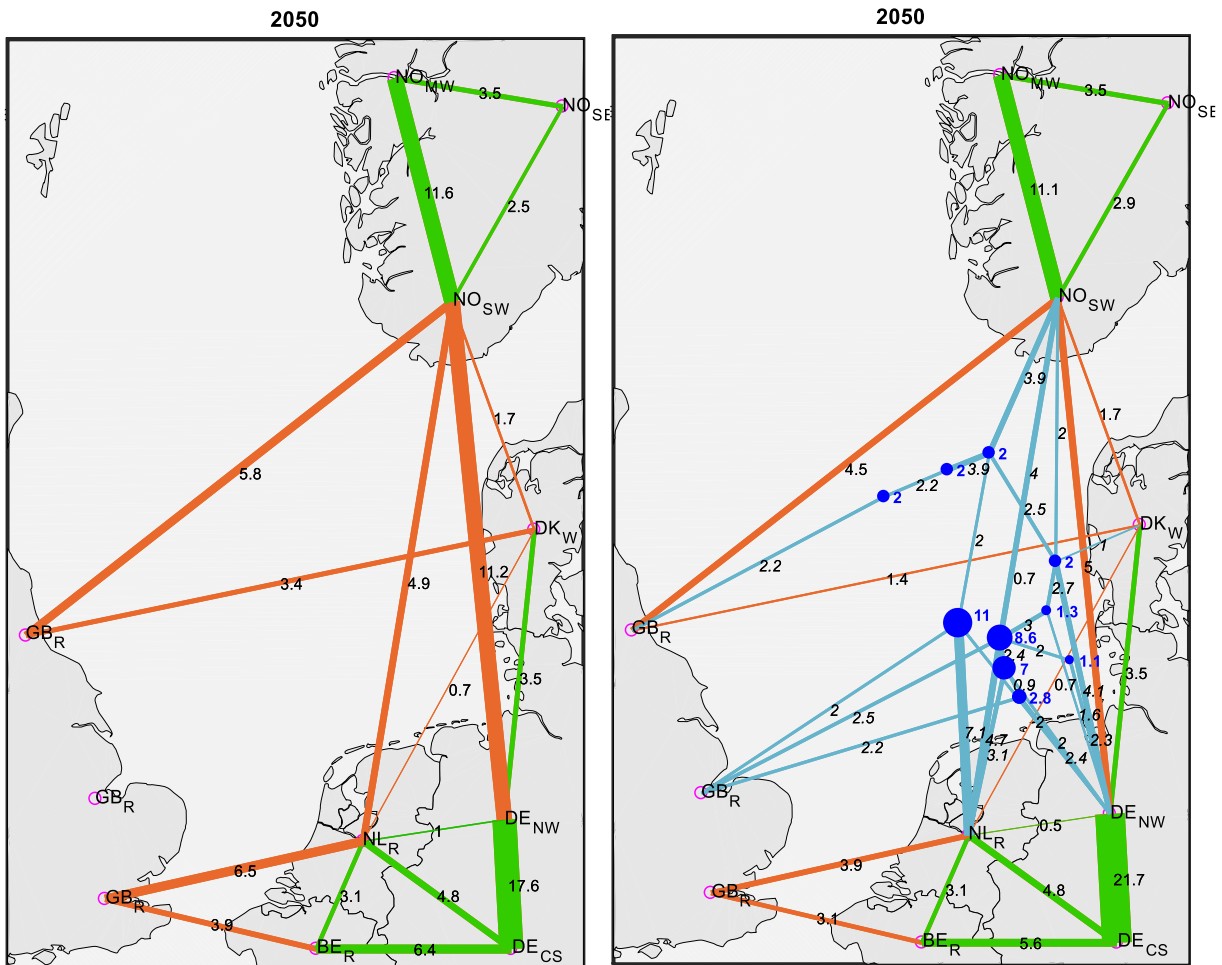

**Figure 3. The resulting transmission and hub-connected offshore wind GW by 2050 in the project-based (left) and offshore grid scenario (right). Green shows on-land lines, orange offshore country-to-country lines, light blue meshed offshore lines and dark blue hub-connected offshore wind power installations. The figures are taken from (Koivisto, Gea-Bermudez, et al., 2019).**

165

### 3.1.1 Renewable energy shares and offshore wind installations

Aggregate results (for countries shown in Figure 2) for the project-based and meshed offshore grid scenarios are shown in Table 2. In both scenarios the renewable generation share increases close to 90 % towards 2050 (in addition to VRE, renewable share includes hydro and biofuels). Total electricity generation remains on 2020 level, as electricity consumption is not

170    changing significantly in these scenarios. Offshore wind installations increase close to 100 GW towards 2050 in the North Sea region, with meshed offshore grid scenario showing 10 GW more offshore wind installations. The meshed offshore grid scenario is also expected to be cheaper than the project-based scenario (Koivisto, Gea-Bermudez, et al., 2019).

**Table 2. Aggregate North Sea region results for the project-based and meshed offshore grid scenarios; data from (Koivisto, Gea-Bermudez, et al., 2019).**

| Scenario | Year | Total electricity generation [TWh] | Renewable generation share in electricity sector (%) | Offshore wind installations [GW] |
|---|---|---|---|---|
| Starting point | Approx. 2020 | 1199 | 46 | 22 |
| Project-based | 2030 | 1188 | 75 | 64 |
| Meshed | | 1193 | 76 | 69 |
| Project-based | 2050 | 1192 | 88 | 92 |
| Meshed | | 1207 | 89 | 102 |

### 3.1.2 VRE curtailment

Table 3 shows VRE curtailment results for the project-based and meshed offshore grid scenarios; data are from (Gea-Bermudez, Das, et al., 2019). Especially in 2050, significant curtailment is expected for wind power. The high share of offshore wind curtailment compared to onshore wind may be a result from Balmorel optimisation; offshore wind is expected to have a higher operational (per MWh) cost than onshore, and thus the curtailment of offshore rather than onshore wind is found optimal in the Balmorel run.

For the analysed region, solar PV curtailment is negligible. It needs to be noted that the reported curtailment considers only high-level transmission grid congestion (between the regions shown in Figure 2), as lower level transmission is not modelled. Thus, there can be additional congestion challenges, especially for generation connected to lower voltage levels, such as rooftop solar PV.

**Table 3. VRE curtailments for the project-based and meshed offshore grid scenarios. Shares with respect to available production.**

| Scenario | Year | Onshore wind (%) | Offshore wind (%) | Total wind (%) |
|---|---|---|---|---|
| Project-based | 2030 | 0.2 | 2.2 | 1.2 |
| Meshed | | 0.2 | 2.4 | 1.3 |
| Project-based | 2050 | 0.2 | 10.3 | 6.0 |
| Meshed | | 0.3 | 10.4 | 6.5 |

### 3.2 Impacts of sector coupling

This section presents the scenario where the modelling of sector coupling has been included in the Balmorel investment optimization.

### 3.2.1 Changes in heat production and electricity load

195 Figure 4 shows how the heating sector is expected to change towards 2050 as an aggregate for the North Sea region countries in focus. Due to electrification, but also as biofuel use is expanded, coal and gas are almost entirely removed from the heating system. Figure 5 shows that electricity load increases as electrification of heating increases and EV fleet expands. It needs to be noted that not all sectors are considered in the presented analysis. Electrification leads to significant increase in required electricity generation, as shown in the next subsection. The possibility to utilise biofuels in the heating sector can be debated;
200 this is discussed more in Section 4.

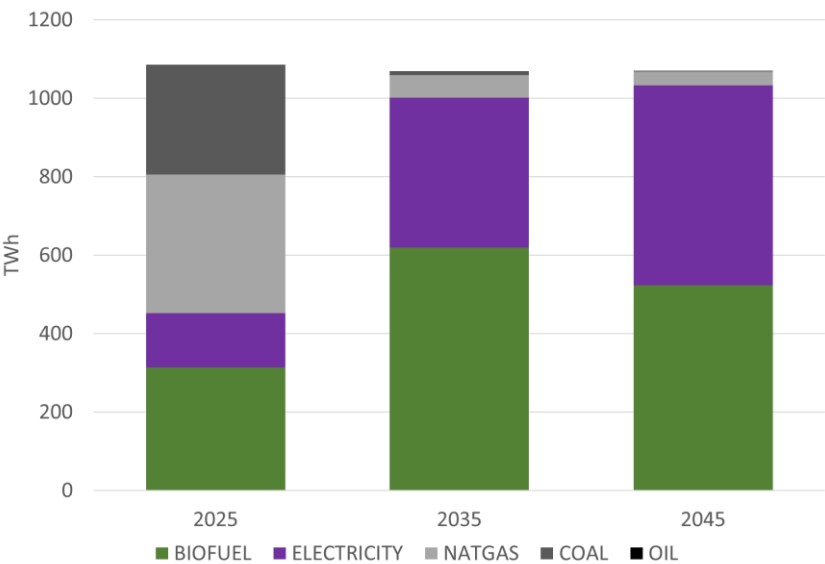

**Figure 4. Aggregated heat production per fuel for the countries in focus (countries shown in Figure 2). The industrial sector and individual users connected to district heating are included.**

205

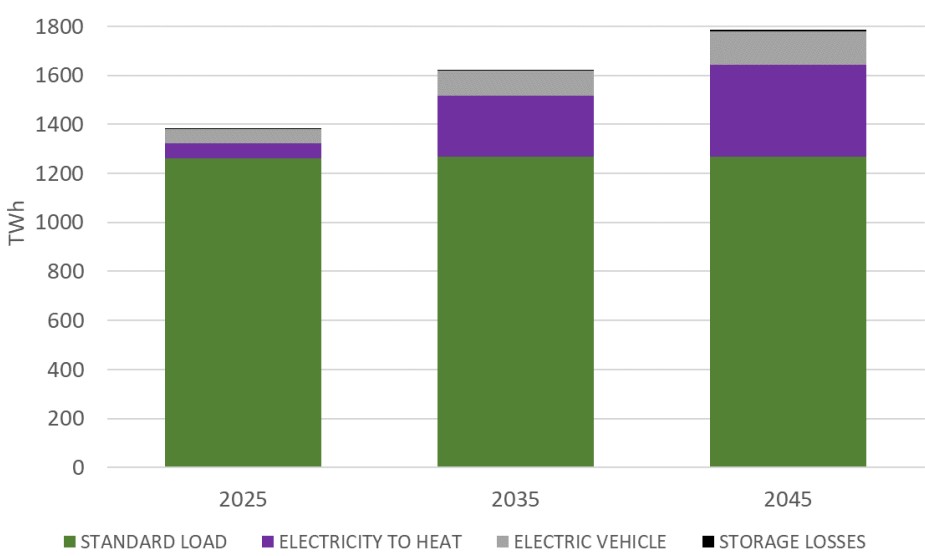

**Figure 5. Aggregated electricity demand per type for the countries in focus. Standard load means classic existing electricity load, with small additional load assumed to come from data centres.**

### 3.2.2 Annual energy generations towards 2050

As can be seen in Table 4, there is a significant increase of electricity generation from the 2020 level shown in Table 2 towards 2045. This can be expected based on the increasing electrification of the heating sector and the expanding EV fleet, and it leads to a significant increase in offshore wind installations. Compared to the impact of meshed offshore grid, as shown in Table 2, the effect of sector coupling is expected to be tens of GW of more of offshore wind power. In addition to increasing the overall level of generation (both GW and TWh), sector coupling increases the renewable generation share from around 90 % in the scenarios presented in Table 2 to close to 100 % in Table 4.

Figure 6 shows how the aggregate annual energy generations from different generation sources develop towards 2050. The system is expected to be highly wind-dominated by 2045, but with some solar generation. The share of offshore wind grows from 2035 to 2045 to cover most of the increased electricity demand, with solar generation also increasing slightly. Some natural gas generation remains until 2045, with hydro and biofuel also providing some dispatchable generation to the system. Storage (other than hydro reservoirs) use increases, especially from 2035 to 2045. This increase is mainly electricity battery storage. By 2025, a significant part of CHP generation takes place in the industry sector. However, after 2025 CHP generation decreases significantly, driven by the increasing electrification of the heat sector.

**Table 4. Aggregate North Sea region results for the countries in focus in the scenario with sector coupling.**

| Year | Total electricity generation [TWh] | Renewable generation share in electricity sector (%) | Offshore wind installations [GW] |
|---|---|---|---|
| 2025 | 1284 | 58 | 25 |
| 2035 | 1537 | 94 | 126 |
| 2045 | 1717 | 96 | 158 |

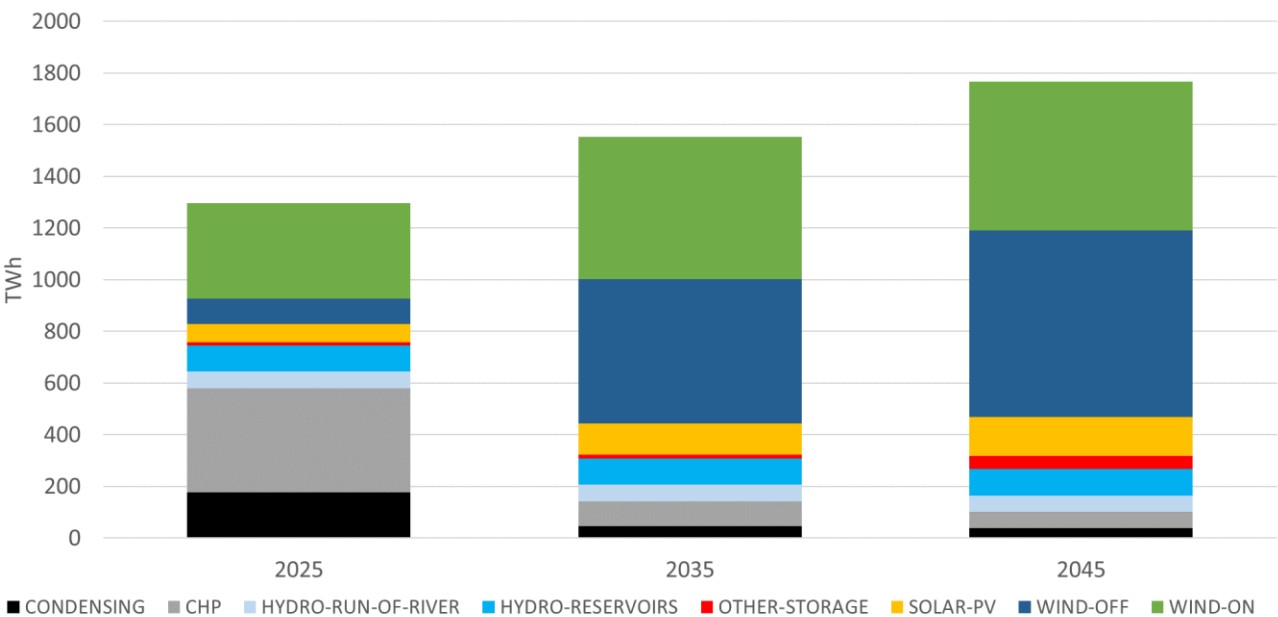

Figure 6. Aggregated annual electricity generation per type for the countries in focus.

### 3.2.3 Offshore wind installations in the different countries

Offshore wind installation development towards 2050 is shown in Table 5. By 2045, almost half of the installed offshore wind generation in the North Sea region countries is expected to be in the UK, driven by large potentials with high CFs. Germany shows the second highest offshore wind capacity development, with Netherlands and Denmark also reaching higher than 10 GW by 2045.

**Table 5. Offshore wind installations per country for the scenario with sector coupling (GW).**

| Country | 2025 | 2035 | 2045 |
|---|---|---|---|
| Belgium | 1.2 | 9.1 | 8.4 |
| Denmark | 2.9 | 5.0 | 11.0 |
| Germany | 9.6 | 43.5 | 46.7 |
| Netherlands | 1.0 | 12.2 | 14.9 |
| Norway | | 3.5 | 4.6 |
| The UK | 10.6 | 52.3 | 72.7 |
| Total | 25.2 | 125.6 | 158.4 |

### 3.2.4 VRE curtailment

VRE curtailment in the scenario with sector coupling is shown in Table 6. The expected level of curtailment is significantly lower than the numbers reported in Table 3. This indicates increased flexibility of the energy system, as it can absorb more VRE generation (Table 4) while simultaneously reducing the curtailment. The specific reasons for this increased flexibility will be studied in future work; however, the strong coupling between the electricity and heating sector (Figure 4) is expected to be a significant contributor. Curtailment of solar PV is negligible.


**Table 6. VRE curtailments for the scenario with sector coupling. Shares with respect to available production.**

| Scenario | Onshore wind (%) | Offshore wind (%) | Total wind (%) |
|---|---|---|---|
| 2025 | 0.0 | 0.0 | 0.0 |
| 2035 | 0.1 | 0.6 | 0.4 |
| 2045 | 0.0 | 1.1 | 0.6 |

### 4 Discussion

The modelled sectors include electricity, district heating, industry and EVs. Individual heating sector modelling and data collection for the analysed countries was not fully completed at the time of writing this paper but will be considered in future 255  research. Ongoing research includes also modelling of the other parts of the transportation sector, such as shipping and aviation, where power-to-gas is expected to play a significantly role.

The $CO_2$ tax levels in 2035 and 2045 may be considered high, although the 2025 level is not far from current market prices of around 25 €/ton in Europe. Future research will consider a $CO_2$ price sensitivity study. In this paper, biofuels play a significant

role in the heating sector. However, when considering, e.g., the transportation sector more broadly, it may be that biofuels are required to cover other demands than the ones modelled in this paper. In addition, considering, e.g., power-to-gas for ships, may change the overall structure of the energy system so that stronger electrification of the heating sector becomes more attractive. These aspects will be considered in future research, where the transportation and heating sectors are covered more comprehensively. $CO_2$ price and other cost sensitivities and sensitivities on assumptions, e.g., on biofuel potentials, can be

used to compare costs of different scenarios.

The impact of assumptions on onshore wind potentials has a significant impact on the offshore wind buildout, with limiting onshore wind driving offshore wind installation (Koivisto & Gea-Bermudez, 2018). Future work will consider how the onshore wind potential assumptions impact offshore wind installations when sector coupling is modelled. Future work will also include

modelling both sector coupling and the meshed offshore grid jointly.

In Table 5, the offshore wind installations in Belgium and Netherlands by 2025 are below the most recent agreed plans in these countries. The starting point for the offshore wind installations will be reassessed in future analyses. When considering offshore wind installations on the level of tens of GW, the impact of large-scale wake losses, where OWPPs cause wake losses to

neighbouring OWPPs (in addition to the internal wake losses inside an OWPP), may become very significant. Future research will consider how to apply such large-scale wake modelling in the context of large-scale energy system analysis.

**5 Conclusion**

This paper has showed that integrating offshore transmission lines and generation investments in the North Sea region can be beneficial and lead to around 10 GW higher offshore wind installation compared to a project-based scenario towards 2050.

Sector coupling is expected to boost offshore wind installations by tens of GW, as electricity consumption increases. In addition, the energy system can benefit from increased flexibility from sector coupling. Indicative results on this were found, as the level of VRE curtailment decreased significantly when sector coupling was considered in the modelling.

**Code and data availability**

The Balmorel model is available at: github.com/balmorelcommunity/Balmorel.

**Author contributions**

Matti Koivisto wrote most of the paper, did the CorRES runs and gave inputs to the Balmorel modelling. Juan Gea-Bermúdez ran the Balmorel optimizations, wrote the Balmorel section of the paper and gave comments. Polyneikis Kanellas helped with

the Balmorel runs and gave comments. Kaushik Das gave inputs to the CorRES and Balmorel runs, especially on the day-by-day runs and curtailment analyses, and provided comments on the paper. Poul Sørensen provided comments.

## Acknowledgements

The authors would like to acknowledge support from the NSON-DK (Danish Energy Agency, EUDP, grant 64018-0032) and PSfuture (DTU Wind Energy) projects.

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
