# Peer review of "North Sea region energy system towards 2050: integrated offshore grid and sector coupling drive offshore wind installations"

_Wind Energy Science, 2020_

## Referee Comment (RC1) · Anonymous Referee #1 · 7 May 2020

This paper looks at three different scenarios for the development of offshore wind in the North Sea region until 2050:

i) Regular electricity and district heating demand and offshore parks connected to country hubs on a project-by-project basis;

ii) Regular electricity and district heating demand and hubs in the North Sea for a meshed grid connecting offshore parks to the mainland;

iii) Adding to the demand industry demand, partial electrification of transport, but no individual heating for buildings.

[Figure]

The paper compares i) and ii) with regards to wind generation and curtailment, then examines these metrics in scenario iii).

The topic is very interesting and the authors have done impactful work in this area in the past, but this paper feels thin and disjointed.

The comparison of i) and ii) repeats results from a previous publication, as the authors admit, adding only the curtailment results, which I don't think is high added value. As they are presented here, the results miss a lot of important information like the total costs, which are presumably in the other paper.

The addition of scenario iii) is an important further development of the model, but I'm not sure why it's bundled together with the comparison of i) and ii) in this paper. Also the results are rather minimal and presented in only 1.5 pages. Surely there is more to say here?

There is so much more that could be considered:

- What is the role of hydrogen? We're seeing a lot of new projects pairing offshore with electrolyzers, which is relevant for industry demand (steel, heat, ammonia, etc.). - Why no heat pumps for individual building heating in France / Germany / UK? Only Scandinavia has significant shares of district heating. - Building renovations? - Wake effects for offshore? DTU has led the field in this analysis. - Dependence of offshore build-out on recent cost developments and low acceptance for onshore wind.

More specific comments:

- The literature review is minimal and mostly contains self-cites. - Figure 1 has low information content. - The year for the reference Gea-Bermudez, Koivisto, & Münster oscillates between 2019 and 2020 or are there 2 publications? - What were the values of the CO2 tax? - Biofuels: what potentials in particular were considered here?

---

## Referee Comment (RC2) · Anonymous Referee #2 · 11 Jun 2020

This paper looks at three different scenarios of developing the North sea offshore wind-farms: 1. Load as of today, offshore wind farms connected to country hubs on a project basis 2. Load as of today, offshore wind farms connected via a meshed grid to the mainland 3. There is a strong sector coupling for the electricity use and windfarm offshore

The different scenarios are compared and the outcome is very clear: a) Meshed grid compared to country connections (1 to 2), no big different b) Sector coupling is very important and make it possible to take away natural gas, coal and oil from the heat production in northern Europe. It also increase the needs of more wind power in the

North Sea.

The results are a very important massage to how the development of the energy supply can develop.

The paper is well written, and it is very easy to catch the main message and it is clear in the results.

The comparison of i) and ii) repeats results from a previous publication, as the authors point out.

The sector coupling is new and very important. Maybe the sector coupling is enough important to be a paper by itself with more background information.

The literature review is limited and there is other paper also describing the sector coupling and its impact for wind energy development, I recommend to make a new review.

---

## Author Comment (AC1) · 9 Aug 2020

Author comment (AC): Response to all referee comments (RCs)

The Authors: We thank the Referees for their constructive comments. The following answers the comments and describes how the paper is revised as a result. As instructed, the revised manuscript will be submitted at a later stage.

Referee #1:

"This paper looks at three different scenarios for the development of offshore wind in the North Sea region until 2050: i) Regular electricity and district heating demand and

offshore parks connected to country hubs on a project-by-project basis; ii) Regular electricity and district heating demand and hubs in the North Sea for a meshed grid connecting offshore parks to the mainland; iii) Adding to the demand industry demand, partial electrification of transport, but no individual heating for buildings. The paper compares i) and ii) with regards to wind generation and curtailment, then examines these metrics in scenario iii). The topic is very interesting and the authors have done impactful work in this area in the past, but this paper feels thin and disjointed. The comparison of i) and ii) repeats results from a previous publication, as the authors admit, adding only the curtailment results, which I don't think is high added value. As they are presented here, the results miss a lot of important information like the total costs, which are presumably in the other paper. The addition of scenario iii) is an important further development of the model, but I'm not sure why it's bundled together with the comparison of i) and ii) in this paper. Also the results are rather minimal and presented in only 1.5 pages. Surely there is more to say here?"

The Authors: We thank the Reviewer for the comments and critique. The presentation of the scenario iii) is expanded in the revised manuscript to show more disaggregated results for the different sectors and countries. For example, the impact of the different sectors on load growth towards 2050 is presented, and the split of offshore wind installations in different countries is shown. Costs of scenarios i) and ii) are presented and compared in the cited papers. Scenario iii) costs are not directly comparable to i) and ii), as additional sectors are included in the analysis; they are thus not compared in this paper. However, different scenarios with sector coupling (with the same sectors modelled, but with different assumptions) are compared in ongoing work. This is mentioned as future research in a Discussion section, which is added to the revised version of the manuscript. The impacts of both the meshed offshore grid and sector coupling on expected offshore wind installations towards 2050 are presented, as both are current popular topics in terms of large-scale energy and power system development in Europe. This paper shows that sector coupling is expected to have a much bigger impact on offshore wind installation growth (which is perhaps not very surpris-

ing as sector coupling drives load growth, which drives VRE installations; however, we considered it to be a worthwhile comparison). Joint modelling of both sector coupling and meshed offshore grid is an ongoing work, which is mentioned in the Discussion section in the revised manuscript.

"There is so much more that could be considered: - What is the role of hydrogen? We're seeing a lot of new projects pairing offshore with electrolyzers, which is relevant for industry demand (steel, heat, ammonia, etc.). - Why no heat pumps for individual building heating in France / Germany / UK? Only Scandinavia has significant shares of district heating. - Building renovations? - Wake effects for offshore? DTU has led the field in this analysis. - Dependence of offshore build-out on recent cost developments and low acceptance for onshore wind."

The Authors: We agree that all these suggestions are very relevant in modelling energy system scenarios. Modelling of hydrogen and synthetic fuel production, also as future fuel in the shipping and aviation sectors, and modelling of electrification of individual building heating are included in current work on expanding the Balmorel model; however, they were not yet ready at the time of writing this paper. They are mentioned as future work in the Discussion section. Large-scale wake modelling at DTU Wind Energy has been recently applied in the North Sea, looking at very large amounts of GW installed in limited geographical regions; however, the combination of this work to the energy system scenario optimisation is also ongoing work, and was not yet available for this paper. It is mentioned as future work. Impact of low acceptance of onshore wind on offshore wind build-out has been looked at in previous work, which is cited in the revised manuscript; however, without considering sector coupling. Doing similar analysis on scenarios with sector coupling is ongoing work.

"More specific comments: - The literature review is minimal and mostly contains self-cites."

The Authors: We agree that the literature review is limited. It is expanded significantly

in the revised version of the manuscript, with focus on papers from other authors.

"- Figure 1 has low information content."

The Authors: The figure is updated in the revised manuscript to include short description of each block.

"- The year for the reference Gea-Bermudez, Koivisto, & Münster oscillates between 2019 and 2020 or are there 2 publications?"

The Authors: The year is 2019, and there is 1 publication only from this group of authors. The citations are corrected in the revised paper.

"- What were the values of the CO2 tax?"

The Authors: Assumed $CO_2$ tax values are: 30, 90 and 120 EUR2015/ton in 2025, 2035, and 2045, respectively. They are the same as in the other related cited papers from the authors (with some interpolations to reach specific scenario years). They are based on the following reference, which is also cited in the revised manuscript: Nordic Energy Research and International Energy Agency, Nordic Energy Technology Perspectives 2016 report (https://www.nordicenergy.org/project/nordic-energy-technology-perspectives/)

"- Biofuels: what potentials in particular were considered here?"

The Authors: The biofuel potentials are the same as used in the following reference (the numbers are presented in the revised version of the manuscript): J. Gea-Bermudez et al.: Optimization of the electricity and heating sectors development in the North Sea region towards 2050, the 18th Int'l Wind Integration Workshop, Dublin, October 2019. (https://backend.orbit.dtu.dk/ws/portalfiles/portal/197181819/Sector_Coupling_Wind_Integration_conference_2019.pdf)

Referee #2:

"This paper looks at three different scenarios of developing the North sea offshore windfarms: 1. Load as of today, offshore wind farms connected to country hubs on
a project basis 2. Load as of today, offshore wind farms connected via a meshed grid to the mainland 3. There is a strong sector coupling for the electricity use and windfarm offshore. The different scenarios are compared and the outcome is very clear: a) Meshed grid compared to country connections (1 to 2), no big different b) Sector coupling is very important and make it possible to take away natural gas, coal and oil from the heat production in northern Europe. It also increase the needs of more wind power in the North Sea. The results are a very important massage to how the development of the energy supply can develop. The paper is well written, and it is very easy to catch the main message and it is clear in the results. The comparison of i) and ii) repeats results from a previous publication, as the authors point out. The sector coupling is new and very important. Maybe the sector coupling is enough important to be a paper by itself with more background information. The literature review is limited and there is other paper also describing the sector coupling and its impact for wind energy development, I recommend to make a new review."

The Authors: We appreciate the comments and critique; they allow us to improve the revised version of the manuscript. The presentation of the scenario with sector coupling (3) is expanded in the revised manuscript to show more disaggregated results for the different sectors and countries. For example, the impact of the different sectors on load growth towards 2050 is presented, and the split of offshore wind installations in different countries is shown. In addition, more background information is given on the assumptions and input data used in the modelling. We agree that the literature review is limited. It is expanded significantly in the revised version of the manuscript.
* * *